# Duplication and Segregation of Centrosomes during Cell Division

**DOI:** 10.3390/cells11152445

**Published:** 2022-08-07

**Authors:** Claude Prigent, Rustem Uzbekov

**Affiliations:** 1Centre de Recherche de Biologie Cellulaire de Montpellier, Université de Montpellier, Centre Nationale de la Recherche Scientific, 34293 Montpellier, France; 2Faculté de Médecine, Université de Tours, 37032 Tours, France; 3Faculty of Bioengineering and Bioinformatics, Moscow State University, 119992 Moscow, Russia

**Keywords:** cell cycle, centrosome, centriole, regulation, checkpoint

## Abstract

During its division the cell must ensure the equal distribution of its genetic material in the two newly created cells, but it must also distribute organelles such as the Golgi apparatus, the mitochondria and the centrosome. DNA, the carrier of heredity, located in the nucleus of the cell, has made it possible to define the main principles that regulate the progression of the cell cycle. The cell cycle, which includes interphase and mitosis, is essentially a nuclear cycle, or a DNA cycle, since the interphase stages names (G1, S, G2) phases are based on processes that occur exclusively with DNA. However, centrosome duplication and segregation are two equally important events for the two new cells that must inherit a single centrosome. The centrosome, long considered the center of the cell, is made up of two small cylinders, the centrioles, made up of microtubules modified to acquire a very high stability. It is the main nucleation center of microtubules in the cell. Apart from a few exceptions, each cell in G1 phase has only one centrosome, consisting in of two centrioles and pericentriolar materials (PCM), which must be duplicated before the cell divides so that the two new cells formed inherit a single centrosome. The centriole is also the origin of the primary cilia, motile cilia and flagella of some cells.

## 1. What Is the Centrosome?

In animal cells, the centrosome is a membrane-less organelle consisting of two interconnected centrioles, PCM and some additional structures [1]. The centriole is a cylinder shape structure with a diameter of about 200 nm and a length of about 500 nm made of nine triplets of microtubules (Figure 1).

The triplets of microtubules are always twisted counterclockwise when viewed from the proximal end of the centriole [2]. Each triplet is composed of microtubules A, B and C, with microtubule C being shorter than the other two and ending in a characteristic hook at the distal end of the centriolar cylinder (Figure 2 middle). Three types of centrioles can simultaneously exist in a cell (Figure 2). The older and most mature centriole (at least 2 cell cycles old) is called the mother centriole in opposition to the younger one, called the daughter centriole (which appeared in the previous cell cycle), and possesses two kinds of appendages named distal and subdistal appendages. Each triplet of microtubules possesses one distal appendage which makes nine distal appendages per centriole [3]. During the formation of the primary cilium distal appendages are essential to attach the centriole to the plasma membrane [4]. The daughter centriole has electron-dense ribs at the site of the future formation of distal appendages (Figure 2, middle).

Regarding subdistal appendages their number varies from 0 to 13 depending on the cell types [5]. The number of subdistal appendages depends on the microtubule nucleating activity of the centrosome, in some types of cells subdistal appendages can be arranged in two or even three rows. The base of each subdistal appendage rests on two or three centriole triplets [3].

Proteins responsible for the nucleation and anchoring of microtubules localize on the heads of the subdistal appendages [6]. These appendages are absent in mitotic centrosomes; they disassemble during G2 phase and reappear on the surface of the mother centriole at the beginning of G1 phase [7].

In the proximal part inner lumen of the centriolar cylinder of the mother and daughter of the centrioles, a system of ligaments is located, which connects the triplets from the inside to each other. In the youngest centriole—procentriole, which arose in the current cell cycle, in the inner lumen there is a “cartwheel structure”, which determines the nine-beam symmetry of centrioles during their formation due to the lateral interaction of SAS6 protein dimers, the angle between which is 40 degrees [8,9].

The centrosome includes not only centrioles, but also several additional structures: satellites, striated rootlets, less morphologically pronounced pericentriolar material and a system of ligaments between the proximal ends of the mother and daughter centrioles. The centrosome is the main nucleation center of microtubules in the cell [10]. Connected to the centrosome are the minus ends of microtubules that attach to the heads of the subdistal appendages, the surface of the predominantly mother centriole and the pericentriolar material (Figure 3).

Finally, the centrosomes nucleate microtubules that participate to four main structures in the cell: the radial MT network in interphase, the bipolar spindle in mitosis, the procentriole and the cilia and flagella [5].

The two centrioles act as a platform to localize and assemble functional protein complexes. For example, the pericentriolar material surrounding the centrosomes consists of proteins that will ensure microtubule nucleation, the main function of centrosomes. Two dynamics of nucleation of microtubules follow each other at the centrosomes during the cell cycle progression. In interphase the nucleated microtubules are not very dynamic and of great length, they will constitute the cytoskeleton of microtubule of the interphase cells and will take part in giving them their shape. In mitosis, the network of interphase microtubules disappears to make room for a much more dynamic network with more numerous and very short microtubules that will only ensure one function: the assembly of the bipolar spindle essential to the segregation of chromosomes. In mitosis, the centrosomes recruit more proteins dedicated to intense nucleation of microtubules, the pericentriolar material of centrosomes becomes more extensive and denser: this mechanism is called the maturation of centrosomes. The mitotic spindle is bipolar, so the cell in mitosis must possess two centrosomes, one for each pole of the spindle. The duplication of centrosomes in interphase must therefore be strictly controlled so that it produces only two centrosomes. A higher number of centrosomes prevents the assembly of a bipolar spindle, disrupts chromosome segregation and induces aneuploidy, a situation frequently observed in cancer cells.

## 2. Common Features between the DNA and the Centrosome Cycle

DNA replication and centrosome duplication during a cell cycle must share common regulations because each of the two new cells formed must inherit only one copy of the genome and one copy of the centrosome from the initial cell [11,12].

The genome of eukaryotic cells is composed of DNA, structured in a double helix that carries the genes. The genome is present in two copies in each cell; each gene is therefore present in the form of two alleles. The cells are therefore diploid. This genome must be replicated (copied) at each cell cycle so that the two daughter cells inherit the same genome as the mother cell. To copy the genome, the double helix is opened and each strand is copied so that each double helix generates two new double helices, each consisting of a mother and a daughter strand [13,14].

The centrosome of higher eukaryotic cells (multicellular organisms) after mitosis consists of two centrioles. The centrosome must be duplicated (copied) during each cell cycle so that the two daughter cells inherit the same centrosome (two centrioles) as the mother cell. During the cell cycle, a new centriole is constructed on the surface of each centriole. Each daughter cell then inherits a centrosome composed of an old centriole (that of the mother cell) and a new one [15].

The temptation was therefore great to consider that, as with DNA, each centriole serves as a model for recreating a new one. However, unlike DNA, which must be copied to preserve genetic sequences and therefore cannot be synthesized de novo, the centriole is not copied. The old centriole only serves as a platform to recruit proteins that will build a new centriole. As evidence, the mechanism can occur de novo in the absence of any centriole [16,17].

The initiation of DNA replication is under the control of the restriction point (R), a checkpoint in G1 phase which when satisfied authorizes the cell to enter the cell cycle which will lead to cell division. As for the DNA, the duplication of the centrosome also depends on this restriction point.

The authorization to enter the cycle results in the production of the E2F family of transcription factors that will transactivate the genes whose products are required either for DNA replication or for centrosome duplication. The start of centrosome duplication precedes DNA replication [18] and the two events are independent of each other. Indeed, the addition of arabinosyl cytosine to cells inhibits DNA replication without affecting centriole duplication [19]. Similarly, centrioles continue to duplicate in sea urchin egg extracts in the presence of aphidicolin, an inhibitor of DNA replication, or even in previously enucleated sea urchin egg extracts [20]. Treatment of CHO cells with hydroxyurea or aphidicolin for a time equivalent to the duration of several cell cycles blocks DNA replication but does not prevent centrioles from continuing to replicate [21].

Another commonality between the DNA and the centrosome is the rule that allows DNA replication and centrosome duplication to occur only once and only once per cell cycle. As replication proceeds, the newly synthesized DNA is no longer allowed to be replicated again, only the passage through a phase of mitosis removes this blockage. The same rule applies to centrosomes which once duplicated are only allowed to replicate once they have passed through mitosis. The modalities of these authorizations are different for the DNA and the centrosome; we will come back to these mechanisms in more detail.

## 3. Behavior of the Centrosome during a Cell Cycle

In G1 phase of the cell cycle the mother centriole with its appendages is connected to the daughter centriole, both centrioles are “disengaged” i.e., allowed to duplicate. The passage of the restriction point will trigger the initiation of their duplication.

This duplication is under the control of the protein kinase PLK4 [22,23,24]. PLK4 comes to localize to each centriole by interacting with the two proteins Cep192 and Cep152 on the centriole wall, initially as a ring around the centriole [25,26,27]. The STIL protein then localizes as a dot on the ring of PLK4 on the centriole wall, STIL recruits Plk4 which relocalizes to the STIL dot, and activates it, in turn Plk4 phosphorylates STIL and this phosphorylated form recruits the SAS6 protein to initiate cartwheel assembly [28,29,30,31,32,33].

The cartwheel, first observed in 1960 in EM images of the base of flagella, is the structural unit of symmetry nine of centrioles [34]. This structure forms even before the nucleation of the microtubules that will form the procentriole. It is from this cartwheel that the growth of the procentriole will start (Figure 4). Once the first cartwheel is formed, others will be made to form a stack of cartwheels that will serve as a scaffold around which the nine triplets of microtubules will be organized.

Our current knowledge of the molecular mechanism of cartwheel establishment is still fragmented and incomplete. The SAS6 protein provides the cartwheel structure [39]. SAS6 forms a homodimer with the N-terminal domains at the center of the cartwheel and the long C-terminal coiled-coil portions of the dimer organize the spokes, 9 dimers organized in a disk form the cartwheel of symmetry 9 (Figure 5).

At the end of each spoke, the STIL protein makes contact with CPAP protein that cooperates with CEP120 to regulate centriolar microtubule growth [40,41,42]. CPAP is essential for procentriole construction [43,44,45] it binds the cartwheel to the nine triplets of microtubules via STIL that might replace by CEP135 during procentriole assembly [46].

Interestingly, CPAP interacts with STIL and/or CEP135, but the function of this interaction is unknown [47]. Both STIL and CEP135 also interact with SAS6. Why and in what cases does SAS6 interact with STIL or CEP135 remains to be determined.

## 4. One and Only One Procentriole per Centriole per Cell Cycle

Each centriole must be duplicated once and only once per cell cycle [48]. What is the mechanism that governs this restriction?

Nucleation of the new procentriole begins on the wall of the old centriole. As discussed above, while PLK4 is initially localized as a ring around the old centriole, a signal breaks this symmetry and induces relocalization of PLK4 to a single site (Figure 5). How is this site defined on the wall of the centriole?

The symmetry breaking of PLK4 localization would be induced by the STIL protein, but this only shifts the problem, how does STIL choose its localization site on the PLK4 ring [49]. Furthermore, it seems that PLK4 is able to self-activate locally and form a focus even in the absence of STIL [50].

In Drosophila syncytial embryos, it was recently observed that the assembly of the new procentriole always starts on the side of the wall that faces the nuclear envelope [51]. The centrosome is indeed always located close to the nucleus and is even physically bound to the nuclear envelope since the centrosome co-purifies with the nuclei during cell fractionations [52].

Could it be that a molecule associated with the nuclear envelope could designate the site of STIL anchoring and/or local PLK4 activation? This remains to be determined.

Obviously, the designation of a single site of construction of the new procentriole on the wall of the old centriole restricts de facto the number of procentrioles made to one per centriole per cycle.

When a new procentriole is built during the S phase of the cycle on the wall of an old centriole, re-duplication is no longer allowed on this centriole. The resulting centrosome is thus formed of a new centriole and old one; both centrioles are considered as “engaged” positioned orthogonally to each other.

### What Are the Proteins Involved in This Engagement?

The first protein identified as essential for this event is CEP215 (or CDK5RAP2 for Cdk5 regulatory subunit associated protein 2), a centriolar protein of the centrosomin family that localizes to the centrosomes during duplication. A deletion of this protein causes an amplification of centrosomes (continuous duplication) due to an engagement defect [53].

It is only when crossing mitosis following duplication that the two centrioles disengage [54]. This event is entirely dependent on the activity of a protease that is activated only when the Spindle Assembly Checkpoint is satisfied, and the cell makes the metaphase to anaphase transition [55,56,57]. This protease degrades cohesin, which binds sister chromatids together allowing their separation, from which it derives its name separase, after having been originally called “sister-separating” factor [58]. The degradation of cohesin thus leads to the segregation of chromosomes. The protease activity of separase is also required to disengage centrioles [57], yet it does not degrade the CEP215 protein [53].

Therefore, there is another protein that ensures centriole engagement and must be degraded by separase to trigger disengagement. This protein is the pericentrin. Indeed, the expression of a pericentrin that cannot be cleaved by separase blocks centriole disengagement and thus duplication [59]. It has been proposed that degradation of pericentrin by separase leads to removal of the CEP215 protein thus eliminating the two proteins ensuring centriole engagement [60].

## 5. Centrosome Splitting, Separation and Segregation

At the end of the duplication process, the two old centrioles are now each engaged with a new centriole. These two old centrioles are still linked by a network of fibrils that must disappear to allow their dissociation and thus form the two new independent centrosomes.

This dissociation of the two old centrioles is under the control of the protein kinase Nek2 (for NIMA-related kinase 2) which phosphorylates the Cep250 protein (CNAP1) to delocalize it [61,62]. This CNAP-1 protein plays a central role, but other proteins also participate in centriole cohesion such as rootletin [63,64], CEP68 and CEP215 [65], conductin [66], LRRC45 [67] and centlein [68].

This process of dissociation of the two centrosomes is essential to allow the separation of the centrosomes, i.e., their migration around the nuclear envelope to a position opposite to each other [69].

The protein that plays a central role in this process is the kinesin-like protein KIF11 (or Kinesin 5). Inhibition of KIF11 by microinjection of antibodies inhibits centrosome separation and induces the formation of monopolar spindle in mitosis [70]. These data were reproduced with the drug monastrol a small molecule that specifically inhibits KIF11 [71]. KIF11 is a motor protein with a motor head domain and a long coiled-coil C terminal domain, forms head-to-head dimers that subsequently dimerized again to form tetramers with two head motors at each end [72]. This tetramer possesses a MT-Sliding activity allowing binding and sliding of two adjacent microtubules [73]. The two centrosomes that are going to separate nucleate microtubules, the microtubules between the two centrosomes will be cross-linked by the KIF11 protein. The MT-sliding activity of KIF11 will result in a push on the centrosomes inducing their separation (Figure 6).

During their separation, the two centrosomes remain connected to the nuclear envelope by binding to dynein, which associates with BICD2 (dynein adaptor) at the nuclear pores [74] or/and the Nup133 protein (a nuclear pore protein) [75].

When the centrosomes split, both centrioles are different; there is a mother centriole and a daughter one. The presence of appendages on a centriole identifies the mother centriole [77]. Information regarding the nature and function of these appendages is still fragmentary and remains to be investigated [78,79]. Unlike distal appendages, subdistal appendages are not present on the mother centriole throughout the cell cycle [80].

Ninein and the dynactin subunit p150 (Glued), for example, are among the proteins involved in the construction of subdistal appendages on the wall of the mother centriole, and p150 (Glued) is required to localize ninein [81]. The p150 (Glued) protein is not present on the mother centriole during mitosis, is released at the end of G2 phase, and reappears in G1 phase. The subdistal appendage disappears from the mother centriole in mitosis during the process of centrosome separation and mitotic spindle assembly. The difference between the two centrioles is then limited to the presence of the distal appendages on the mother centriole [7].

The function of the subdistal appendage is to anchor microtubules during interphase [6], the reason why it is no longer present in mitosis remains to be determined. This could be related to the change in dynamics of microtubule nucleation by centrosomes and thus to their maturation.

## 6. Conclusions

Eventually, it is not so surprising that the centrosome cycle is a complex and strictly regulated process. The cell goes through various stages to duplicate its centrosome and segregate each of the new one in the two newly generated cells that must inherit one and only one centrosome. The different stages of the cell cycle G1, S, G2 and M have been defined in reference to the cycle of the DNA, consequently, it is not always an easy task to describe the centrosome cycle using the phases of the DNA cycle (Figure 7).

However, the comparison with DNA (the genome) is not without interest because as each somatic cell in G1 should contain one and only one copy of its diploid genome, it must also contain one and only one centrosome. This proportion is maintained after each cell division. Chromosome segregation during mitosis is performed by a dynamic spindle of microtubules and in order for the mother cell to separate two sets of chromosomes for the two newly generated cells, the spindle must be bipolar.

This bipolarity is controlled by the number centrosomes, one at each pole. Any changes in the centrosome number affects spindle assembly and consequently cell ploidy. There are few exceptions such the megakaryocyte that is polyploid and contains multiple centrosomes, in this cell the proportion is conserved two copies of the diploid genome with two centrosomes, 4 with 4, 8 with 8, 16 with 16 and so on. This is probably because abortive mitoses are at the origin of this polyploidy and centrosome amplification [82]. However, the more surprising behavior of megakaryocytes is that they still assemble multiple bipolar spindle using a mechanism that is not fully understood [83].

In agreement with Boveri’s theory many cancer cells are aneuploid and possess more than two centrosomes [84,85,86]. However, one of the properties of cancer cells is their ability to adapt, which often poses problems for treatment. For instance, cancer cells have found a way to deal with multiple centrosomes and still assemble a bipolar spindle even with more than two centrosomes. The mechanism named centrosome clustering consists by being able to form a bipolar spindle with one centrosome at one pole and three clustered centrosomes at the other pole [87]. Even if the mechanism is not fully understood, an inhibition of the clustering mechanism has been shown to be effective in eliminating cancer cells that have developed this strategy to proliferate in the presence of supernumerary centrosomes [88].

It is then established that some cells can assemble a bipolar spindle with different number of centrosomes at each pole, but what would happen in the absence of centrosome? Are the centrosomes essential to assemble a bipolar spindle? The answer is clearly “No” for three main reasons; (1) meiotic bipolar spindle in mouse oocytes is assembled in the absence of centrosomes [17], (2) bipolar spindle can assemble after laser ablation of centrosomes [89] and (3) Drosophila *PLK4* mutants assemble bipolar mitotic spindle without centrosome [90]. So, cells can assemble bipolar spindles without centrosomes and cells can generate a centriole de novo. Then why do somatic cells keep conserving a centrosome? Moreover, why does the canonical mechanism imply that the new centriole is assembled on the surface of another one?

The presence of the centrosome might be necessary to limit duplication: each centriole giving birth to only one centriole. Indeed, de novo centriole amplification can be obtained by overexpressing PLK4 only after having removed the existing centrosome implying that its presence is limiting duplication [91].

Although the centrosome was discovered almost 150 years ago [92,93,94], the mechanism of duplication as well as its regulation during cell cycle progression are still not fully understood. Its conservation during evolution and its involvement in so many functions in the cell should attract more young researchers. To borrow a phrase from Michel Bornens, who has passed on his passion for the centrosome to many of us: “there are many important questions raised by this organelle” [95].

## Figures and Tables

**Figure 1 cells-11-02445-f001:**
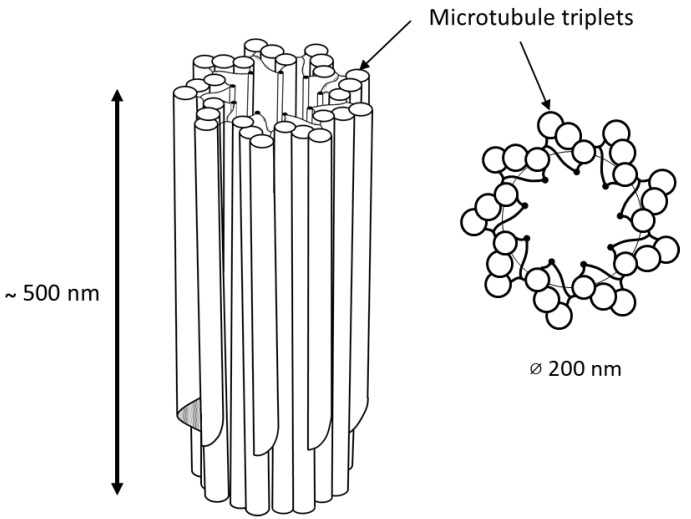
9 triplets of microtubules to form a centriole.

**Figure 2 cells-11-02445-f002:**
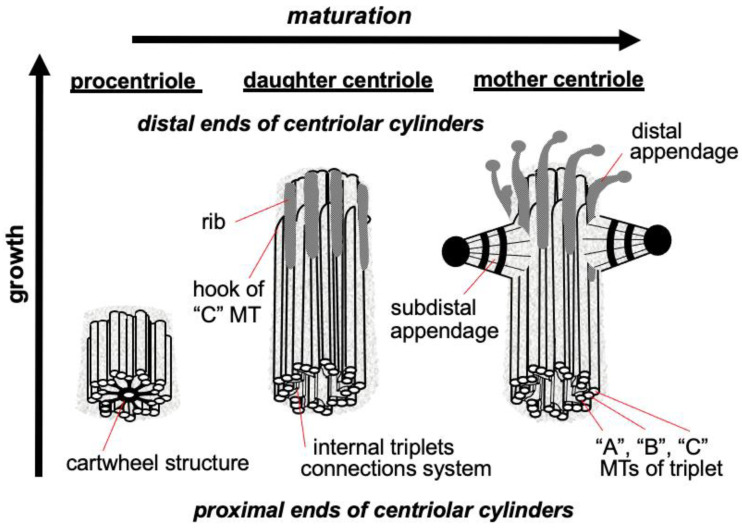
Procentriole, daughter and mother centrioles. Three morphological types of centriolar cylinders that can simultaneously exist in a cell. The mother centriole has an “age” at least 2 cell cycles, the daughter centriole originated in the previous cell cycle, the procentriole appeared in the current cell cycle.

**Figure 3 cells-11-02445-f003:**
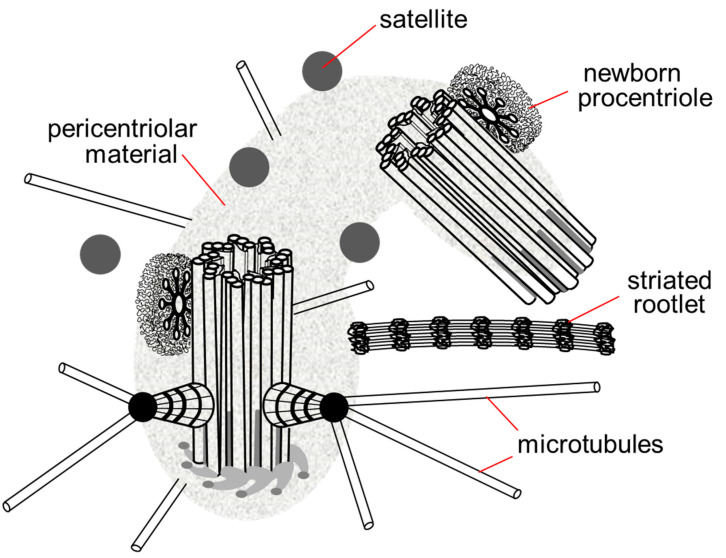
Centrosome in the late G1 phase of the cell cycle. The pericentriolar material completely surrounds the mother centriole and the proximal end of the daughter centriole, the satellites are located around the centrosome, in contact with the pericentriolar material, the striated rootlet is not an obligatory component of the centrosome in non-ciliary cells, its functions are not known.

**Figure 4 cells-11-02445-f004:**
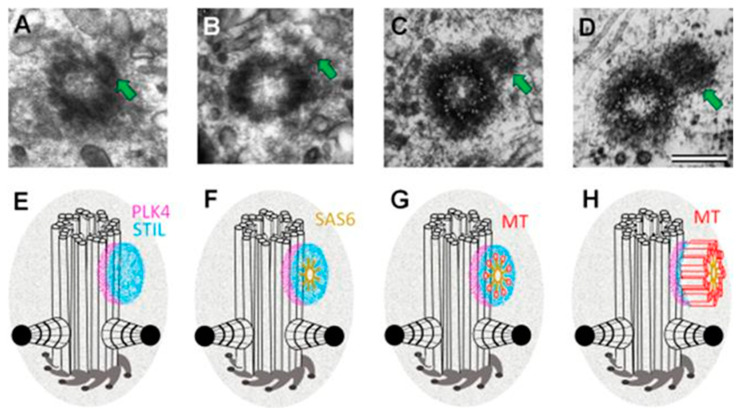
Ultrastructure of the early stages of centriole formation on the mother centriole in the G1 phase of the cell cycle ((**A**,**B**)—from [18,35,36] with modifications) and localization of the proteins involved in this process (**E**,**H**) after [32,37]). The photographs show cells of the porcine embryonic kidney line (SPEV) fixed after mitosis at a time significantly less than the average (**B**) or even minimum (**A**,**C**,**D**) duration of the G1 phase of the cell cycle for this cell line [38]. (**A**,**E**)—at the stage of a dense disc, a complex of PLK4 kinase and STIL protein is formed on the centriole surface. (**B**,**F**)—at the stage of the removed disk, 9 dimers of the SAS6 protein are added to this complex, which, due to the interaction of their lateral domains, form the “cartwheel structure”. (**C**,**G**)—nucleation of centriole microtubules occurs at the ends of the “cartwheel structure” spokes; the figure shows MT singlets. (**D**,**H**)—MT singlets are completed to MT doublets and further to MT triplets. The length of procentrioles at the end of the G1 phase of the cell cycle is about 100 nm; further elongation of triplets is activated when the cell enters the S phase of the cell cycle. On the surface of the mother centriole, the schemes show 9 distal appendages and two subdistal appendages (such a number of subdistal appendages is typical for this cell line). Microtubules extending from the heads of subdistal appendages and pericentriolar material are not shown in the diagrams. Arrows show procentrioles. The photo and diagrams show only one (mother) centriole from a pair. It should be noted that the mother centriole in the diagrams is shown when viewed from the proximal end and the triplets are twisted counterclockwise at this point of view, the procentriole is shown when viewed from the distal end, so the triplets in the diagram are twisted clockwise [2]. Scale bar 200 nm.

**Figure 5 cells-11-02445-f005:**
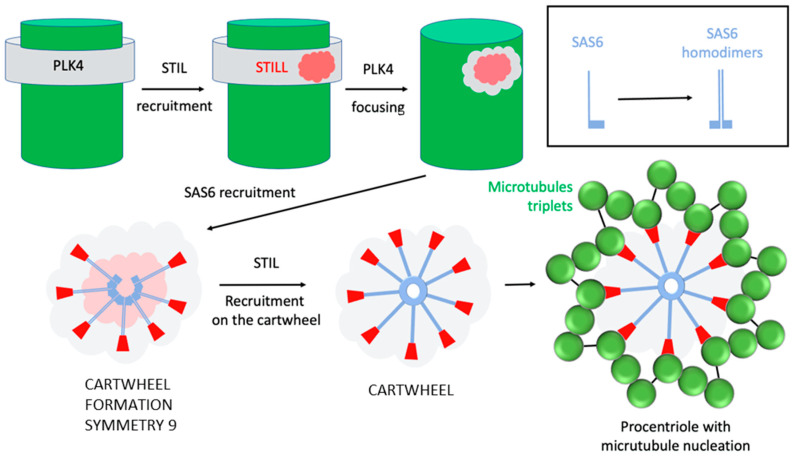
Cartwheel assembly: PLK4 (**grey**), located as a ring around the centriole, relocates by focusing on the STIL protein (**pink**). The assembly of the cartwheel begins with 9 SAS6 doublets stabilized by STIL, at the extremity of the coiled-coil regions of each doublet. STIL will ensure the link with each triplet of microtubules of the centrioles.

**Figure 6 cells-11-02445-f006:**
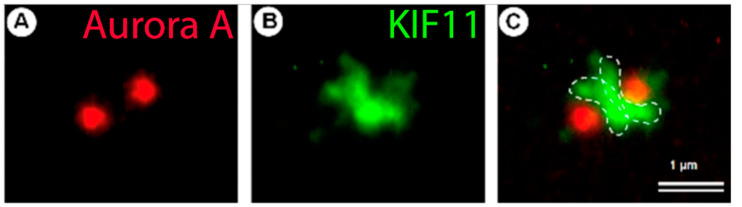
Kinesin-like motor KIF11 participate in centrosome separation in G2 phase of cell cycle. (**A**) Centrosomes labeled with monoclonal antibodies against Aurora A are shown in red, (**B**) localization of KIF11 between two centrosomes at the beginning its separation was detected with labelling by polyclonal antibodies against KIF11 in green; (**C**) double labelling show superposition. The dotted lines show the probable binding regions of the KIF11 protein to microtubules extending from two centrosomes. Scale bar 1 µm. Photos made by R. Uzbekov from [76] with modifications.

**Figure 7 cells-11-02445-f007:**
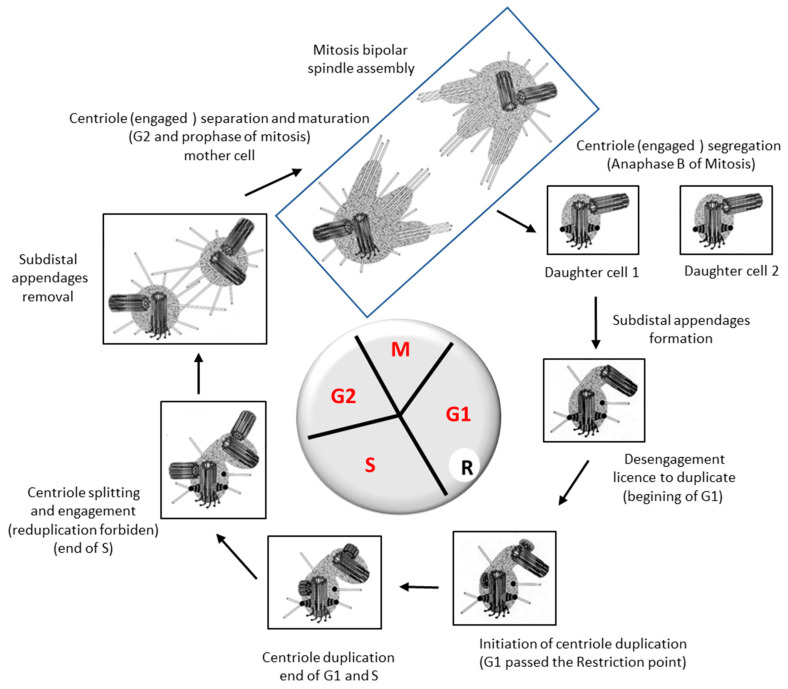
Stages of the centriolar cycle and their relationship with the stage of the cellular (nuclear) cycle.

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
