# Peer review of "Duplication and Segregation of Centrosomes during Cell Division"

_cells, 2022, doi:10.3390/cells11152445_

Round 1

Reviewer 1 Report

This review from Prigent and Uzbekov focuses on regulation of centrosome cycle during cell division. This is a very interesting paper that help to understand the molecular mechanism regulating centriole duplication. Moreover, the Authors correlate the DNA with the centrosome cycle and very clear schemes have been inserted which definitely improve the paper. Unfortunately there are some aspects that need to be improved before pubblication. My main observation concerns the description of centrosome and centrioles. I note that sometimes it is not very clear the difference between centrosome and centrioles, especially in the first part of the paper. The Authors should check this out.

Here my consideration:

Abstract:

Pag1 line 5: The Authors should change this sentence because cell cycle does not identify DNA cycle only. Cell cycle comprises interphase (G1, S, G2) and mitosis, Authors must clarify to avoid misunderstandings

Pag 1, Line 9: Centrosome consists of 2 centrioles and pericentriolar materials (PCM) from which microtubules originate. Authors should change this sentence.

Pa. 1 line 12 Centriole (not centrosome), acting as basal body plays a central role in organizing axoneme of cilium and flagellum.

Authors should clarify

Pag. 1 line 14: “without discussing ciliogenesis " in the opinion of this referee it is not necessary to specify this aspect. Authors should eliminate this sentence.

Pag 2 line 2: the Authors should report that centrosome is formed by two centrioles and PCM

Pag 2 line 7: Only two types of centrioles and not three are described here. Authors should clarify.

Pag.3 line 4: Authors should changes this sentence:

The appendages are present during interphase, they disassemble before mitosis and reappear on mother centrioles at the end of mitosis.

Pag. 10 line 18: Rotletin not rotlelin

Pag 11 Fig. 6. If the Authors did not perform this immunolocalization, References for this picture should be indicated

Pag. 14 line 4: Organ or organelle?

In conclusion this review is very interesting, however, there are some questions, as reported above, that should be addressed before publication.

I would like to stress that the paper needs some re-writing in the description of centrosome and centrioles that sometimes are not clearly distinguishable in their description.

Author Response

This review from Prigent and Uzbekov focuses on regulation of centrosome cycle during cell division. This is a very interesting paper that help to understand the molecular mechanism regulating centriole duplication. Moreover, the Authors correlate the DNA with the centrosome cycle and very clear schemes have been inserted which definitely improve the paper. Unfortunately there are some aspects that need to be improved before pubblication. My main observation concerns the description of centrosome and centrioles. I note that sometimes it is not very clear the difference between centrosome and centrioles, especially in the first part of the paper. The Authors should check this out.

Here my consideration:

We thank the reviewer for the appreciation of our work. Below we answered all the questions of the reviewer and indicated the changes we made to the text of the article.

Abstract:

Pag1 line 5: The Authors should change this sentence because cell cycle does not identify DNA cycle only. Cell cycle comprises interphase and mitosis, Authors must clarify to avoid misunderstandings

The DNA cycle has thus become known as the cell cycle (G1/S/G2/M), the S phase corresponding to the DNA synthesis phase.

We have changed the phrase in the abstract to be more specific about why cell cycle names are actually nuclear cycle names. The names "gap 1" and "gap 2" themselves speak for themselves. After all, in fact, in these “intervals” important events of the cell cycle occur, which are not reflected in their names in any way. This is exactly what we wanted to emphasize.

The cell cycle, which includes interphase and mitosis, is essentially a nuclear cycle, or a DNA cycle, since the interphase stages names (G1, S, G2) phases are based on processes that occur exclusively with DNA.

 Pag 1, Line 9: Centrosome consists of 2 centrioles and pericentriolar materials (PCM) from which microtubules originate. Authors should change this sentence.

We did not find this sentence in the paper. May be reviewer wanted to add this sentence? We added it in the abstract.

Pag. 1 line 12 Centriole (not centrosome), acting as basal body plays a central role in organizing axoneme of cilium and flagellum. Authors should clarify

Corrected.

Pag. 1 line 14: “without discussing ciliogenesis " in the opinion of this referee it is not necessary to specify this aspect. Authors should eliminate this sentence.

Eliminated

Pag 2 line 2: the Authors should report that centrosome is formed by two centrioles and PCM

Corrected.

Pag 2 line 7: Only two types of centrioles and not three are described here. Authors should clarify.

Procentriole, daughter centriole and mother centriole have different structure, so we present it on the figure 2. In S phase these three different types clearly visible, because procentriole at this time significantly smaller, and mother in differ from daughter have distal and subdistal appendages.

Pag.3 line 4: Authors should changes this sentence:

The appendages are present during interphase, they disassemble before mitosis and reappear on mother centrioles at the end of mitosis.

Corrected.

Pag. 10 line 18: Rotletin not rotlelin

rootletin [63,64], in the text now

Pag 11 Fig. 6. If the Authors did not perform this immunolocalization, References for this picture should be indicated

It is photo prepared by Rustem Uzbekov from Giet et al., 1999 with modifications. We added reference to the paper.

Pag. 14 line 4: Organ or organelle?

The quote from [96] has been corrected.

Reviewer 2 Report

The manuscript reviewed the “Duplication and segregation of centrosomes during cell division”. The paper is of interest, but some points must be considered prior acceptance.

Overall, the work is well written and organized, although there are some typing errors to correct during the revision of the work.

Please use the template for manuscript preparation.

Authors are recommended to start the content with Introduction, and followed by each sub-topic.

There are many abbreviations in the manuscript. It is suggested that authors incorporate a List of Abbreviation in the manuscript.

Author Response

The manuscript reviewed the “Duplication and segregation of centrosomes during cell division”. The paper is of interest, but some points must be considered prior acceptance.

Overall, the work is well written and organized, although there are some typing errors to correct during the revision of the work.

Please use the template for manuscript preparation.

Authors are recommended to start the content with Introduction, and followed by each sub-topic.

There are many abbreviations in the manuscript. It is suggested that authors incorporate a List of Abbreviation in the manuscript.

Thank you for your offer. We have added a list of abbreviations at the end of the article. Noticed typing errors have been corrected.

List of Abbreviations:

BICD2 (dynein adaptor) - Bicaudal-D2 (BICD2) is a dynein activating adaptor protein that plays a critical role in microtubule-based minus-end-directed transport. 

CHO cells - Chinese hamster ovary cells

G1 phase (from Gap 1) - phase after mitosis before DNA replication

G2 phase (from Gap 2) – phase of cell cycle after S pahse and before mitosis.

CEP68 - Centrosomal Protein 68kDa

CEP120 - Centrosomal Protein 120kDa

CEP135 - Centrosomal Protein 135kDa

Cep152 - Centrosomal Protein 152kDa

Cep192 - Centrosomal Protein 192kDa

CEP215 - Centrosomal Protein 215kDa (or CDK5RAP2 for Cdk5 regulatory subunit associated protein 2)

CNAP1 (Centrosomal Protein 250kDa - Cep250) centrosomal Nek2-associated protein 1

CPAP - centriolar protein (centrosomal P4.1-associated protein)

EM – electron microscopy

LRRC45 - Leucine Rich Repeat Containing 45 kDa

Kinesin-like protein KIF11 (or Kinesin 5 or XlEg5) – plus end directed microtubule associated motor.

M phase – mitosis   

MT - microtubule

Nek2 -  for NIMA-related kinase 2

Nup133 protein -  nuclear pore protein 133kDa

p150(Glued) protein - Dynactin subunit, potent anti-catastrophe factor for microtubules PCM - pericentriolar material

PLK4 – Polo-like kinase 4 (Serine/threonine-protein kinase)

S phase (from synthesis) – phase of cell cycle during which DNA replicated

SAS6 - Centriolar Assembly Protein (Spindle Assembly 6 Homolog from Caenorabditis Elegans)

STIL protein - Centriolar Assembly Protein, the SCL/TAL interrupting locus is a human oncogene that was originally identified from human T-cell lymphoblastic leukemia, where it functions as a hematopoietic transcription factor.

Reviewer 3 Report

This is a review manuscript in which the authors described the centrosome behavior during cell division. In this manuscript the authors used highly condensed description telling a whole story of how cetrosomes are composed, organized and developed from lots of peer works during the nearly 150 years since the centrosomes were discovered, showing a wide knowledge and deep understanding within this field of study. I think it is ready to be published. Some minor correction may happen when the subtittles 3,4 and 5 should be aligned with other subtitle to the beginning of the line.

Author Response

This is a review manuscript in which the authors described the centrosome behavior during cell division. In this manuscript the authors used highly condensed description telling a whole story of how cetrosomes are composed, organized and developed from lots of peer works during the nearly 150 years since the centrosomes were discovered, showing a wide knowledge and deep understanding within this field of study. I think it is ready to be published. Some minor correction may happen when the subtittles 3,4 and 5 should be aligned with other subtitle to the beginning of the line.

We thank the reviewer for the high evaluation of our work. We have aligned subtittles 3, 4 and 5.

Round 2

Reviewer 1 Report

This reviiewer appreciate a lot that Authors answered all questions reporting the changes they made to their article. I retain that this review is now ready for pubblication.

Just a little changes could be made in the abstract:

Pag 1 line 13 after centrosome a comma could be added and I suggest to correct the sentence in this way:

……….centrosome, consisting in of 2 centrioles and pericentriolar materials (PCM),……….

Finally, this referee greatly appreciates the dedication of this review to Michel Bornens, a point of reference for all those who work on the centrosome.